# Hand and Oral Hygiene Practices among Adolescents in Dominican Republic, Suriname and Trinidad and Tobago: Prevalence, Health, Risk Behavior, Mental Health and Protective Factors

**DOI:** 10.3390/ijerph17217860

**Published:** 2020-10-27

**Authors:** Supa Pengpid, Karl Peltzer

**Affiliations:** 1ASEAN Institute for Health Development, Mahidol University, Salaya 73170, Thailand; supaprom@yahoo.com; 2Department of Research Administration and Development, University of Limpopo, Sovenga 0727, South Africa; 3Department of Psychology, University of the Free State, Bloemfontein 9300, South Africa

**Keywords:** oral hygiene, hand hygiene, school adolescents, correlates, Caribbean

## Abstract

Objective: The study aimed to estimate the prevalence and correlates of oral hygiene (OH) and hand hygiene (HH) behavior among school adolescents in three Caribbean countries. Method: In all, 7476 school adolescents (median age 14 years) from the Dominican Republic, Suriname, and Trinidad and Tobago responded to the cross-sectional Global School-Based Student Health Survey (GSHS) in 2016–2017. Results: The prevalence of poor OH (tooth brushing < 2 times/day) was 16.9%, poor HH (not always before meals) was 68.2%, poor HH (not always after toilet) was 28.4%, and poor HH (not always with soap) was 52.7%. In the adjusted logistic regression analysis, current cannabis use, inadequate fruit and vegetable intake, poor mental health, and low parental support increased the odds for poor OH. Rarely or sometimes experiencing hunger, trouble from alcohol use, inadequate fruit and vegetable intake, poor mental health, and low parental support were associated with poor HH (before meals and/or after the toilet, and/or with soap). Conclusion: The survey showed poor OH and HH behavior practices. Several sociodemographic factors, health risk behaviors, poor mental health, and low parental support were associated with poor OH and/or HH behavior that can assist with tailoring OH and HH health promotion.

## 1. Introduction

Oral hygiene (OH) (“tooth brushing ≥ 2/day”) is a major tool to prevent and control periodontal diseases and dental caries [1]. Good hand hygiene (HH) using soap can “avert 0.5–1.4 million deaths every year” [2]. Despite the potential positive impact of good OH and HH, the prevalence of good OH and HH practices among adolescents is low [3,4,5,6,7,8]. There is a lack of recent national information on OH and HH among adolescents in Caribbean countries, such as the Dominican Republic, Suriname, and Trinidad and Tobago [4]. Among school adolescents in 15 Latin American and Caribbean countries from 2006–2011, 2–9% reported infrequent (<one time/day) tooth brushing and 2–7% infrequent (never or rarely) HH after toilet use [4]. In the Dominican Republic, “risk factors for diarrhea and cholera transmission include poor adherence to water, sanitation, and hygiene (WASH) practices such as consistent hand washing” [9,10,11]. In Suriname, the prevalence of caries in schoolchildren was moderate to high (using WHO criteria), and the majority of children had dental caries [12]. The prevalence of tooth-cleaning (≥2 times/day) among adults in the Dominican Republic was 94.2% [13].

In a multi-country investigation among school adolescents, most of the respondents (80%) reported daily tooth brushing, and more than one in 20 students brushed their teeth “less than once a day or never” in half of the countries [5], including 10.0% in Zambia [14]. In a study among school-going adolescents in nine African countries, the results showed that 22.7% had poor OH (tooth brushing < 2/day), 62.2% had poor HH (not always before meals), 58.4% had poor HH (not always after toilet), and 35.0% had poor HH (not always with soap) [15], whereas in six Southeast Asian countries, 17.1% had poor OH, 44.8% had poor HH (before meals), 31.9% had poor HH (after toilet), and 55.8% had poor HH (with soap) [3].

As reviewed earlier [3], “factors associated with poor OH among adolescents include sociodemographic variables (early adolescence, male sex, and lower wealth status), health risk behaviours (insufficient fruit and vegetable consumption, physical inactivity, and tobacco use), poor mental health and lack of parental support”. Factors associated with poor HH among adolescents [3], include “male sex, lower wealth status, health risk behaviours (low fruit and vegetable consumption, sedentary behaviour and physical inactivity, and substance use), poor mental health, and lack of parental support”. 

Recent national data on the prevalence and correlates of OH and HH among adolescents are lacking in the Caribbean region. Therefore, the aim of this study was to assess the prevalence of OH and HH in three Caribbean countries. We looked at sociodemographic factors, health risk behaviors, and protective factors of poor OH and HH among adolescents in the Dominican Republic, Suriname, and Trinidad and Tobago in 2016–2017.

## 2. Methods

### 2.1. Sample and Procedures

Cross-sectional nationally representative survey data from the 2016 Dominican Republic, 2016 Suriname, and 2017 Trinidad and Tobago Global School-Based Student Health Survey (GSHS) were analyzed [16]. The sampling approach included a two-stage sampling design, including schools and classes. All school students present in a selected classroom were eligible to participate by filling in a self-administered anonymous questionnaire [16]. Information that was more detailed can be publically accessed [16]; the overall response rate was 63% in the Dominican Republic, 83% in Suriname, and 89% in Trinidad and Tobago [16]. Ethical approval was obtained from the ethics committees in each study country, and participants gave written consent [16].

### 2.2. Measures

The GSHS questions used [16] are described in Appendix A.

Oral hygiene was sourced from the question, “During the past 30 days, how many times per day did you usually clean or brush your teeth?” Responses ranged from “1 = I did not clean or brush my teeth during the past 30 days to 4 = 6 or more times per day (coded 1–3 = 1 and 4–6 = 0)”.

Hand hygiene was sourced from three items. “During the past 30 days, how often did you wash your hands before eating/after using the toilet or latrine/use soap?” Response options ranged from “1 = never to 5 = always (coded 1–4 = 1 and 5 = 0)”. 

Sociodemographic information included country, sex, age, and experiences of hunger (as a proxy for economic status).

Health risk behaviors assessed included current tobacco use, trouble from alcohol use, current cannabis use, fruit and vegetable consumption, leisure-time sedentary behavior, physical activity, and attendance of physical education classes.

Poor mental health was defined as “0 = 0 low, 1 = 1 medium, and 2–5 = 2 high based on positive responses to the items no close friends, loneliness, anxiety, suicidal ideation and suicide attempt” [3]. *Protective factors* included school attendance, peer, and parental support. “The four items on parental or guardian support were summed and classified into three groups, 0–1 low, 2 medium, and 3–4 high support” [3].

### 2.3. Data Analysis

Statistical analyses were conducted with STATA software version 15 (Stata Corporation, College Station, TX, USA), taking into account the sampling weights and multistage sampling design. Multivariable logistic regression analyses were used to predict the covariates of poor OH (tooth brushing <2/day), poor HH (not always before meals), poor HH (not always after toilet), and poor HH (not always with soap). Missing variable information was excluded from the analysis and *p*-values < 0.05 indicated statistical significance. 

## 3. Results

### 3.1. Sample Characteristics and Hygiene Behavior

The study sample comprised 7476 school adolescents (14 years median age; interquartile range = 13–16) from the Dominican Republic, Suriname and Trinidad and Tobago. In all three countries, the prevalence of poor OH (<twice a day tooth brushing) was 16.9%, poor HH (not always before meals) was 68.2%, poor HH (not always after toilet) was 28.4%, and poor HH (not always with soap) was 52.7% (see Table 1).

### 3.2. Associations with Poor Oral and Hand Hygiene 

In adjusted logistic regression analysis, students from Trinidad and Tobago (adjusted odds ratio (AOR): 1.63, 95% confidence interval (CI): 1.23–2.14), current cannabis use (AOR: 2.41, 95% CI: 1.31–4.45), inadequate fruit and vegetable intake (AOR: 2.24, 95% CI: 1.02–4.93), and high poor mental health (AOR: 1.39, 95% CI: 1.00–1.91) increased the odds, and high parental support (AOR: 0.70, 95% CI: 0.51–0.97) decreased the odds for poor oral hygiene. 

Compared to students from the Dominican Republic, students from Suriname and Trinidad and Tobago had lower odds for poor HH before meals and after the toilet, but higher odds for using soap. The male sex had lower odds for poor HH after the toilet, and older adolescents had higher odds for poor HH with soap. Rarely or sometimes experiencing hunger was positively, and high parental support was negatively associated with all three poor HH indicators (before meals, after the toilet, and with soap). Inadequate fruit and vegetable intake and moderate peer support were associated with poor HH after meals and with soap, whereas moderate or high poor mental health increased the odds for poor HH after meals and after toilet. Trouble from alcohol use was positively associated with poor HH with soap, and physical inactivity and sedentary behavior were negatively associated with poor HH after toilet (see Table 2).

## 4. Discussion

This study gives an insight on important health preventive behaviors such as OH and HH in three Caribbean countries. The prevalence of poor OH (16.9%) seemed similar in six countries in Southeast Asia (17.1%) [3], lower in African countries (22.7%) [15], and lower in three countries in Oceania (22–38%) [17]. The prevalence of poor HH (before meals) (68.2%) was higher in nine countries in Africa (37.8%) [15], six Southeast Asian countries (44.8%) [3], and Pacific island states (30% to 35%) [17]. Poor HH (after toilet) (28.4%) was similar to the Southeast Asian study (31.9%) [3], but lower than in nine African countries (41.6%) [15]. Poor HH (with soap) (52.7%) was lower than in African countries (65.0%) [15], but similar to Southeast Asian countries (55.8%) [3]. 

Poor OH was significantly higher in Trinidad and Tobago (22.1%) than in the Dominican Republic (16.2%), poor HH (before meals) was the highest among adolescents in the Dominican Republic, and poor HH (after toilet) was the highest in the Dominican Republic (30.7%), whereas poor HH (with soap) was the highest in Trinidad and Tobago (60.3%) in this study. The higher prevalence of poor HH (after toilet) in the Dominican Republic may be attributed to the lowest access to improved water sources and sanitation facilities (86%), compared to Suriname (93%), and Trinidad and Tobago (94%) [18]. The prevalence of poor HH (with soap) was the highest in Trinidad and Tobago (60.3%), followed by Suriname (56.2%), and the Dominican Republic (51.2%), whereas based on national household surveys, the “percentage of households with a specific place for hand washing where water and soap or other cleansing agent are present” was the highest in Trinidad and Tobago (94.6%) [19], followed by Suriname (80.1%) [20], and the Dominican Republic (56.1%) [21]. The national coverage estimate for water and sanitation in schools was 80% for water and 65% for sanitation in Suriname, and 100% for water and 100% for sanitation in Trinidad and Tobago [22]. It is possible that poorer HH among adolescents in Trinidad and Tobago is related to the poor implementation of the Health and Family Life Education (HFLE) school health program, which includes personal hygiene [23].

While some previous research [3,5,15,24,25] showed a positive association between the male sex and poor OH and/or poor HH, this survey did not find significant sex differences, apart from the male sex being negatively associated with poor HH (after the toilet). In a study among adolescents in Zambia, the male sex was negatively associated with poor OH [11]. Compared to students who were never hungry, students who were rarely or sometimes hungry had higher odds for poor HH (before meals, after the toilet, and with soap). This result was in line with former studies [7,14,17,26,27] as it showed an association between lower economic status (experience of hunger) and poor OH and/or poor HH. It is possible that adolescents from poorer households have less access to toothbrushes and/or soap. 

Consistent with previous studies [3,6,8,14,15,17,28], this survey showed a positive association between health risk behaviors (current cannabis use, trouble from alcohol use, and inadequate fruit and vegetable intake), poor mental health, and lack of parental support with poor OH and/or poor HH (before meals, and/or after the toilet, and/or with soap). “It is possible that fruit and vegetable consumption, which is known to have positive effects on well-being, acts as mediators in the correlation between poor mental health and personal hygiene” [8]. Contrary to results from previous studies [3,15,17,24], physical inactivity and sedentary behavior were negatively associated with poor HH (after the toilet), and moderate peer support was associated with poor HH (before meals and with soap). Despite this, other health risk behaviors (current cannabis use, trouble from alcohol use, and inadequate fruit and vegetable intake) seemed to cluster with poor OH and/or poor HH [3,4,26,29]. Health promotion programs should promote “hand-washing with soap and tooth brushing with tooth paste,” together with the identified clustering health risk behaviors and poor mental health, to prevent dental conditions and infectious diseases, such as diarrhea and cholera, in Caribbean countries [2,9,10,11,30,31]. 

## 5. Limitations 

The study focused only on school adolescents and was cross-sectional in design. Furthermore, the self-report of the data, including OH and HH, could have biased responses. Yet, self-reported measures of OH have been used as a proxy tool for reporting indicators of clinical OH among adolescents [32]. The difficulty of comparing the prevalence rates of poor HH is that different definitions have been used, e.g., never or rarely (rather than sometimes, most of the time, or always) washing hands after toilet use [4,8], and not always (never, rarely, sometimes, or most of the time) washing hands after toilet use [3,15,17,24].

## 6. Conclusions

This study, including three national surveys among school adolescents in the Caribbean in 2016–2017, showed a high proportion of poor OH (<twice a day tooth brushing) (16.9%), poor HH (not always before meals) (68.2%), poor HH (not always after toilet) (28.4%), and poor HH (not always with soap) (52.7%). Several sociodemographic factors, health risk behaviors, poor mental health, and low parental support were associated with poor OH and/or poor HH behavior that can assist with tailoring OH and HH health promotion.

## Figures and Tables

**Table 1 ijerph-17-07860-t001:** Characteristics of the sample and hygiene behavior in the Dominican Republic, Suriname, and Trinidad and Tobago, 2016–2017.

Variable	Sample	Not Always Hand Hygiene			Oral Hygiene
		Before Meals	After Toilet	With Soap	Tooth Brushing < 2/day
	N (%)	%	%	%	%
All	7476	68.2	28.4	52.7	16.9
Sociodemographic variables					
Country					
Dominican Republic	1481 (19.8)	69.9	30.7	51.2	16.2
Suriname	2126 (28.4)	46.0	14.1	56.2	14.3
Trinidad and Tobago	3869 (51.8)	66.6	19.8	60.3	22.1
Gender					
Female	3916 (50.6)	68.1	28.0	52.2	14.0
Male	3466 (49.4)	67.5	26.3	51.7	16.8
Age in years					
≤13	2210 (29.7)	56.9	22.8	50.1	17.0
14–15	3148 (42.3)	67.5	26.1	47.4	14.9
≥16	2083 (28.0)	71.8	31.4	57.3	17.8
Went hungry					
Never	4011 (62.2)	64.4	23.6	48.4	15.2
Rarely/sometimes	2802 (33.9)	75.2	35.9	60.0	17.9
Mostly/always	589 (3.9)	67.2	31.6	53.5	26.0
Health risk behavior					
Current tobacco use	910 (12.1)	73.9	44.3	59.4	31.2
Trouble from alcohol use	661 (12.1)	71.8	41.5	69.0	25.1
Current cannabis use	365 (4.4)	67.5	42.6	62.1	32.1
Fruit/Vegetables (<5 servings/day)	5756 (82.4)	70.5	28.2	54.5	17.6
Sedentary behavior	3354 (47.4)	71.3	27.8	55.4	18.5
Physical inactivity	5920 (86.6)	68.3	28.5	53.1	17.5
No physical education	2117 (27.3)	72.7	33.4	58.2	21.5
Poor mental health					
Low	4200 (64.4)	64.7	23.1	50.3	12.9
Moderate	1440 (18.4)	71.1	32.1	55.0	20.2
High	1309 (17.3)	74.1	37.3	55.8	19.7
Protective factors					
School attendance	5658 (74.9)	67.6	25.9	50.9	15.4
Peer support					
Low	2305 (32.3)	67.8	34.7	55.7	20.6
Moderate	2067 (29.0)	74.8	35.8	61.9	21.9
High	2756 (38.7)	64.9	20.0	47.2	11.5
Parental support					
Low	1811 (35.5)	78.8	37.1	64.1	22.2
Medium	2179 (27.2)	68.0	28.7	56.7	18.1
High	6782 (37.3)	58.3	16.7	40.6	9.2
Parents not using tobacco	5506 (85.3)	67.8	27.5	51.5	15.3

**Table 2 ijerph-17-07860-t002:** Associations with poor oral hygiene (OH) and poor hand hygiene (HH) indicators.

Variable	Poor OH	Poor HH (before Meals)	Poor HH (after Toilet)	Poor HH (with Soap)
	AOR (CI 95%)	AOR (CI 95%)	AOR (CI 95%)	AOR (CI 95%)
Sociodemographic variables				
Country				
Dominican Republic	1 (Reference)	1 (Reference)	1 (Reference)	1 (Reference)
Suriname	1.09 (0.72, 1.64)	0.31 (0.25, 0.37) ***	0.29 (0.20, 0.42) ***	1.34 (1.13, 1.59) ***
Trinidad and Tobago	1.63 (1.23, 2.14) ***	0.77 (0.61, 0.97) *	0.41 (0.31, 0.55) ***	1.43 (1.13, 1.82) **
Gender				
Female	1 (Reference)	1 (Reference)	1 (Reference)	1 (Reference)
Male	0.95 (0.66, 1.37)	0.81 (0.54, 1.21)	0.76 (0.58, 0.98) *	0.83 (0.63, 1.09)
Age in years				
≤13	1 (Reference)	1 (Reference)	1 (Reference)	1 (Reference)
14–15	1.10 (0.69, 1.73)	1.38 (0.98, 1.93)	1.07 (0.78, 1.47)	0.96 (0.69, 1.34)
≥16	0.95 (0.71, 1.26)	1.57 (1.00, 2.47)	1.21 (0.75, 1.97)	1.57 (1.04, 2.38) *
Went hungry				
Never	1 (Reference)	1 (Reference)	1 (Reference)	1 (Reference)
Rarely/sometimes	1.11 (0.80, 1.52)	1.51 (1.25, 1.84) ***	2.04 (1.67, 2.49) ***	1.40 (1.02, 1.93) *
Mostly/always	1.72 (0.85, 3.49)	0.94 (0.57, 1.54)	1.75 (0.73, 4.19)	1.35 (0.76, 2.41)
Health risk behavior				
Current tobacco use	0.83 (0.56, 1.23)	0.89 (0.52, 1.55)	0.81 (0.95, 2.49)	0.73 (0.46, 1.16)
Trouble from alcohol use	1.09 (0.55, 2.19)	1.03 (0.61, 1.75)	1.31, 0.78, 2.20)	2.05 (1.17, 3.25) *
Current cannabis use	2.41 (1.31, 4.45) **	0.60 (0.24, 1.52)	1.05 (0.24, 4.57)	1.25 (0.42, 3.69)
Fruit/Vegetables (<5 servings/day)	2.24 (1.02, 4.93) *	1.69 (1.27, 2.25) ***	1.18 (0.83, 1.67)	1.53 (1.01, 2.32) *
Sedentary behavior	0.89 (0.62, 1.28)	1.07 (0.87, 1.32)	0.62 (0.42, 0.91) *	1.03 (0.85, 1.26)
Physical inactivity	1.12 (0.86, 1.48)	0.73 (0.49, 1.10)	0.65 (0.51, 0.82) ***	0.81 (0.52, 1.27)
No physical education	1.11 (0.77, 1.62)	1.10 (0.72, 1.68)	0.98 (0.66, 1.47)	1.02 (0.64, 1.61)
Poor mental health				
Low	1 (Reference)	1 (Reference)	1 (Reference)	1 (Reference)
Moderate	1.22 (0.82, 1.83)	1.38 (1.03, 1.86) *	1.51 (0.89, 2.57)	1.00 (0.73, 1.36)
High	1.39 (1.00, 1.91) *	1.37 (0.95, 1.97)	1.69 (1.30, 2.18) ***	0.87 (0.59, 1.29)
Protective factors				
School attendance	0.84 (0.47, 1.50)	1.10 (0.85, 1.43)	0.81 (0.61, 1.07)	0.89 (0.76, 1.05)
Peer support				
Low	1 (Reference)	1 (Reference)	1 (Reference)	1 (Reference)
Moderate	0.97 (0.72, 1.32)	1.73 (1.27, 2.34) ***	1.13 (0.69, 1.85)	1.47 (1.01, 2.15) *
High	0.79 (0.42, 1.49)	1.39 (0.90, 2.13)	0.87 (0.55, 1.38)	1.17 (0.81, 1.68)
Parental support				
Low	1 (Reference)	1 (Reference)	1 (Reference)	1 (Reference)
Medium	1.19 (0.81, 1.52)	0.61 (0.39, 0.95) *	0.91 (0.68, 1.21)	0.94 (0.62, 1.43)
High	0.70 (0.51, 0.97) *	0.37 (0.28, 0.50) ***	0.49 (0.33, 0.73) ***	0.51 (0.37, 0.70) ***
Parents not using tobacco	1.10 (0.60, 2.02)	0.76 (0.55, 1.04)	0.91 (0.65, 1.28)	0.88 (0.71, 1.09)

AOR: adjusted odds ratio; CI: confidence interval; *** *p* < 0.001; ** *p* < 0.01; * *p* < 0.05.

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
