# Peer review of "Hand and Oral Hygiene Practices among Adolescents in Dominican Republic, Suriname and Trinidad and Tobago: Prevalence, Health, Risk Behavior, Mental Health and Protective Factors"

_ijerph, 2020, doi:10.3390/ijerph17217860_

Round 1

Reviewer 1 Report

To the authors,

For consideration: Keywords as Oral Hygiene and Hand hygiene could be added for better search possiblity.

Introduction

Page 1

Suggestion for improving: Paragraph one, line 11, sentence starting with "In Suriname, the prevalence of caries in schoolchildren ... and the majority of schoolchildren..."

Paragraph two, line one, first sentence, consider adding "... most of the respondents (80%) reported daily tootbrushing..."

Page 2

line 1, consider to clarify "... in nine African countries the results showed that 22.7% had poor OH.... line two missing the Word Had "58.4% had poor HH" and "35.0% had poor HH"; same in line 4 "31.9% hade poor HH... "55.8% had poor HH"

Paragraph 2, line 2-3 suggestion "...Health risk behaviours (insuficient fruit and vegetable....and Tobacco use) and Line 5-6 same suggestion to put example in brackets: " Health risk behaviours (low fruit...subtance abuse)"

Page 2 last paragraph. last line, last sentence missing of after prevanlence " In all Three countries, the prevalence of poor OH..."

Page 3, line 2 missing "was" Before 28.4%

Discussion

Paragraph one, first sentence, consider revision of the statement: "This study show the newest results...." for example "This study gives an insight on important Health preventive behaviours such as OH and HH..."

Paragraph one, second sentence consider adding to the statment "lower than in African countries " to "but was lower than ..." to be more clear.

Paragraph one, third sentence, consider revise to "...in nine African countries"

Paragraph 4 and 5 layout to be revised, sentence has been devided. Paragraph 5, sentence starting "it is possible that ..." there is a mention of "in Trinidad and Tobago" two times, consider cross out the one at the end of the sentence.

Page 6, paragraph one, line 8 the sentence that starts with "Programmes" should be clarified, suggestion "Health promotion programms should promote..."

References

The layout should be revised as there is a lot of underline text like in ref 1; ref 10, ref 13, ref 21 and 22; ref 27, 28, 29 and ref 31 and 32.

Layout for ref 7.

Good Luck with the revision!

Author Response

Reviewer 1:

For consideration: Keywords as Oral Hygiene and Hand hygiene could be added for better search possiblity.
Response: added
Introduction
Page 1
Suggestion for improving: Paragraph one, line 11, sentence starting with "In Suriname, the prevalence of caries in schoolchildren ... and the majority of schoolchildren..."
Response: Corrected
Paragraph two, line one, first sentence, consider adding "... most of the respondents (80%) reported daily tootbrushing..."
Response: added
Page 2
line 1, consider to clarify "... in nine African countries the results showed that 22.7% had poor OH.... line two missing the Word Had "58.4% had poor HH" and "35.0% had poor HH"; same in line 4 "31.9% hade poor HH... "55.8% had poor HH"
Response: Corrected
Paragraph 2, line 2-3 suggestion "...Health risk behaviours (insuficient fruit and vegetable....and Tobacco use) and Line 5-6 same suggestion to put example in brackets: " Health risk behaviours (low fruit...subtance abuse)"
Response: Corrected

Page 2 last paragraph. last line, last sentence missing of after prevanlence " In all Three countries, the prevalence of poor OH..."
Response: Corrected
Page 3, line 2 missing "was" Before 28.4%
Response: Corrected

Discussion
Paragraph one, first sentence, consider revision of the statement: "This study show the newest results...." for example "This study gives an insight on important Health preventive behaviours such as OH and HH..."
Response: Corrected

Paragraph one, second sentence consider adding to the statment "lower than in African countries " to "but was lower than ..." to be more clear.
Response: Corrected
Paragraph one, third sentence, consider revise to "...in nine African countries"
Response: Corrected

Paragraph 4 and 5 layout to be revised, sentence has been devided. Paragraph 5, sentence starting "it is possible that ..." there is a mention of "in Trinidad and Tobago" two times, consider cross out the one at the end of the sentence.
Response: Corrected

Page 6, paragraph one, line 8 the sentence that starts with "Programmes" should be clarified, suggestion "Health promotion programms should promote..."
Response: added
References
The layout should be revised as there is a lot of underline text like in ref 1; ref 10, ref 13, ref 21 and 22; ref 27, 28, 29 and ref 31 and 32.
Layout for ref 7.
Response: Corrected

Health promotion programms should promote..."
Response: Corrected

Good Luck with the revision!

Reviewer 2 Report

The manuscript submitted to IJERPH entitled “Prevalence and associated factors of hand and oral hygiene behaviour among adolescents in Dominican Republic, Suriname and Trinidad and Tobago” is an original article which aim to estimate the prevalence and correlates of oral hygiene (OH) and hand hygiene (HH) behaviour among school adolescents in three Caribbean countries.

On my opinion the article is interesting, well written, with good English. The content of the manuscript is very interesting. The authors showed poor OH and HH behaviour practices.

However, I highlighted some issues.

Article Type: Is "Brief Report" correct?

Abstract. It may be helpful to structure the abstract to attract the reader's attention.

Introduction. Are there studies in the literature concerning diseases in the student population? Better specify the objectives and methods of the study.

Methods. Is it possible to insert the link of a webpage or a copy of the questionnaire as supplementary material? Specify, if possible, the number of approvals of the ethics committees.

Discussion. Are there other similar studies that have shown similar results in the adult population? Did the authors find limitations in their study by comparing it with other in the literature?

I don't feel qualified to judge about the English language and style.

There is no conflict of interest between me and any of the authors.

Author Response

Reviewer 2:
Comments and Suggestions for Authors
The manuscript submitted to IJERPH entitled “Prevalence and associated factors of hand and oral hygiene behaviour among adolescents in Dominican Republic, Suriname and Trinidad and Tobago” is an original article which aim to estimate the prevalence and correlates of oral hygiene (OH) and hand hygiene (HH) behaviour among school adolescents in three Caribbean countries.
On my opinion the article is interesting, well written, with good English. The content of the manuscript is very interesting. The authors showed poor OH and HH behaviour practices.
However, I highlighted some issues.
Article Type: Is "Brief Report" correct?
Response: Corrected to article

Abstract. It may be helpful to structure the abstract to attract the reader's attention.
Response: changed to structured abstract
Introduction. Are there studies in the literature concerning diseases in the student population?
Response: below is reported
In the Dominican Republic, “risk factors for diarrhea and cholera transmission include poor adherence to water, sanitation, and hygiene (WASH) practices such as consistent hand washing” [9-11]. In Suriname, the prevalence of caries of schoolchildren was moderate to high (using WHO criteria), and the majority of children had dental caries [12].
Better specify the objectives and methods of the study.
Response: various additions were made
Methods. Is it possible to insert the link of a webpage or a copy of the questionnaire as supplementary material?
Response: the GSHS questionnaire used in this study is added as supplementary File 1
Specify, if possible, the number of approvals of the ethics committees.
Response: below is added
Ethical approval was obtained from ethics committees at each study country, and participants gave written consent [16].

Discussion. Are there other similar studies that have shown similar results in the adult population?
Response: below is reported on the adult population
The prevalence of teeth cleaning (≥2 day) among adults in the Dominican Republic was 94.2% [13].

Did the authors find limitations in their study by comparing it with other in the literature?
Response: below is added
A difficulty of comparing prevalence rates of poor HH is that different definitions have been used, e.g., never or rarely (rather than sometimes, most of the time, or always) washing hands after toilet use [4,8], and not always (never, rarely, sometimes or most of the time) washing hands after toilet use [3,15,17,24].

I don't feel qualified to judge about the English language and style.
There is no conflict of interest between me and any of the authors.

Reviewer 3 Report

The aim of this study was to evaluate the prevalence and factors of hand and oral hygiene behavior and establish correlations with other health risk behavior in adolescents from the Dominican Republic, Suriname and Trinidad and Tobago in 2016-2017. The data of the work could be interesting for a specific population, although the authors indicated that an ethical approval is not necessary for the present analysis, the editorial team should assess this, since these data are extracted from other institutions. In contrast, the authors have taken this line of research previously successfully (Reference 3, 12, 21)

I suggest reconsidering the title of the work and include “Health risk behavior”, since the work is also focused on this.

In my opinion, the discussion has been poor. I consider that it would need more foundation, considering sociodemographic aspects of the specific population, in order to clarify and give a possible explanation (if any) to the different correlations presented in the work, also contributing what advantages or disadvantages these could have results in public health. Although the current discussion is a good basis, I believe that it should be strengthened considering these aspects.

Specific comments:

Reference 3 refers to Asian countries. If this reference cannot be replaced by a global one, it must be specified in the introduction that said prevalence belongs to a specific population.

Page 2 (Results - Sample characteristics and hygiene behavior). The abbreviation IQR must be defined.

Author Response

The aim of this study was to evaluate the prevalence and factors of hand and oral hygiene behavior and establish correlations with other health risk behavior in adolescents from the Dominican Republic, Suriname and Trinidad and Tobago in 2016-2017. The data of the work could be interesting for a specific population, although the authors indicated that an ethical approval is not necessary for the present analysis, the editorial team should assess this, since these data are extracted from other institutions. In contrast, the authors have taken this line of research previously successfully (Reference 3, 12, 21)
Response: Below is added
Ethical approval was obtained from ethics committees at each study country, and participants gave written consent [16].
I suggest reconsidering the title of the work and include “Health risk behavior”, since the work is also focused on this.
Response: changed to below
Hand and oral hygiene practices among adolescents in Dominican Republic, Suriname and Trinidad and Tobago: Prevalence, health risk behaviour, mental health and protective factors

In my opinion, the discussion has been poor. I consider that it would need more foundation, considering sociodemographic aspects of the specific population, in order to clarify and give a possible explanation (if any) to the different correlations presented in the work, also contributing what advantages or disadvantages these could have results in public health. Although the current discussion is a good basis, I believe that it should be strengthened considering these aspects.
Response: as below

reviewer 3 is unclear about this point.
all issues have been discussed, e.g., the sociodemographic aspects are discussed as below, including giving possible explanations

While some previous research [3,5,15,24,25] showed a positive association between male sex and poor OH and/or poor HH, this survey did not find significant sex differences, apart from male sex being negatively associated with poor HH (after toilet). In a study among adolescents in Zambia, male sex was negatively associated with poor OH [11]. Compared to students who were never hungry, students who were rarely or sometimes hungry had higher odds for poor HH (before meals, after toilet, and with soap). This result is in line with former studies [7,14,17,26,27], showing an association between lower economic status (experience of hunger) and poor OH and/or poor HH. It is possible that adolescents from poorer households have less access to tooth brushes and/or soap.

Specific comments:
Reference 3 refers to Asian countries. If this reference cannot be replaced by a global one, it must be specified in the introduction that said prevalence belongs to a specific population.
Response: changed to below
Despite the potential positive impact of good OH and HH, the prevalence of good OH and HH practices among adolescents is low [3-8].
Page 2 (Results - Sample characteristics and hygiene behavior). The abbreviation IQR must be defined.
Response: Corrected

Round 2

Reviewer 3 Report

The authors have corrected the manuscript according to my comments. They have clarified a point that I did not interpret correctly, after this clarification I consider that the article would be suitable for publication. It only has to be taken into account, in the edition of the manuscript, that in the title there is a comma between "health, risk behavior" that should be removed.